

# Genetic and phylogenetic analysis of Chinese sacbrood virus isolates from *Apis mellifera*

Ming Li, Dongliang Fei, Li Sun* and Mingxiao Ma*

College Animal Husbandry and Veterinary, Jinzhou Medical University, Jinzhou, Liaoning, China
* These authors contributed equally to this work.

## ABSTRACT

**Background:** Sacbrood virus (SBV) is one of the most pathogenic honeybee viruses that exhibits host specificity and regional variations. The SBV strains that infect the Chinese honeybee *Apis cerana* are called Chinese SBVs (CSBVs).

**Methods:** In this study, a CSBV strain named AmCSBV-SDLY-2016 (GenBank accession No. MG733283) infecting *A. mellifera* was identified by electron microscopy, its protein composition was analyzed by sodium dodecyl sulfate polyacrylamide gel electrophoresis and agar gel immunodiffusion assay, and its nucleotide sequence was identified using a series of reverse-transcription polymerase chain reaction fragments of AmCSBV-SDLY-2016 generated using SBV/CSBV-specific primers. To investigate phylogenetic relationships of the CSBV isolates, a phylogenetic tree of the complete open reading frames (ORF) of the CSBV sequences was constructed using MEGA 6.0; then, the similarity and recombination events among the isolated CSBV strains were analyzed using SimPlot and RDP4 software, respectively.

**Results:** Sequencing results revealed the complete 8,794-nucleotide long complete genomic RNA of the strain, with a single large ORF (189–8,717) encoding 2,843 amino acids. Comparison of the deduced amino acid sequence with the SBV/CSBV reference sequences deposited in the GenBank database identified helicase, protease, and RNA-dependent RNA polymerase domains; the structural genes were located at the 5′ end, whereas the non-structural genes were found at the 3′ end. Multiple sequence alignment showed that AmCSBV-SDLY-2016 had a 17-amino acid (aa) and a single aa deletion at positions 711–729 and 2,128, respectively, as compared with CSBV-GD-2002, and a 16-aa deletion (positions 711–713 and 715–728) as compared with AmSBV-UK-2000. However, AmCSBV-SDLY-2016 was similar to the CSBV-JLCBS-2014 strain, which infects *A. cerana*. AmCSBV-SDLY-2016 ORF shared 92.4–97.1% identity with the genomes of other CSBV strains (94.5–97.7% identity for deduced amino acids). AmCSBV-SDLY-2016 was least similar (89.5–90.4% identity) to other SBVs but showed maximum similarity with the previously reported CSBV-FZ-2014 strain. The phylogenetic tree constructed from AmCSBV-SDLY-2016 and 43 previously reported SBV/CSBV sequences indicated that SBV/CSBV strains clustered according to the host species and country of origin; AmCSBV-SDLY-2016 clustered with other previously reported Chinese and Asian strains (AC genotype SBV, as these strains originated from *A. cerana*) but was separate from the SBV genomes originating from Europe (AM genotype SBV,

Corresponding authors
Li Sun, lilybaby80@163.com
Mingxiao Ma, lnjzmmx@163.com

originating from *A. mellifera*). A SimPlot graph of SBV genomes confirmed the high variability, especially between the AC genotype SBV and AM genotype SBV. This genomic diversity may reflect the adaptation of SBV to specific hosts, ability of CSBV to cross the species barrier, and the spatial distances that separate CSBVs from other SBVs.

# INTRODUCTION

Sacbrood virus (SBV) is one of the most pathogenic honeybee viruses that infects larvae and causes larval death. SBV is also known to infect adult bees and reduce the honey production capacity. At present, SBV has been globally found (*Allen & Ball, 1996*; *Berenyi et al., 2006*; *Ellis & Munn, 2005*). SBV is a single positive-stranded picornavirus in the genus *Iflavirus* that encodes one large open reading frame (ORF), with structural genes located at the 5′ end and nonstructural genes at the 3′ end (*Chen, Evans & Feldlaufer, 2006*). The complete genomic sequence of AmSBV-UK-2000 (in this study, we refer to the previously reported SBV/CSBV strains as "SBV/CSBV + isolated geographic location + year") was first determined by *Ghosh et al. (1999)*.

The SBV strains that infect the Asian honeybee *Apis cerana* are called Chinese SBVs (CSBVs). There is no distinct difference between the nonstructural proteins of SBV and CSBV; however, the structural protein VP1 has the highest variation in amino acid sequence among the proteins from different SBV strains (*Cheng et al., 2011*). SBV has been divided into two major groups—AC genotype SBV originating from *A. cerana* and AM genotype SBV from *A. mellifera*—based on the SBV VP1 sequence (*Ma et al., 2013b*). The AC genotype may be divided further into subgroups. The differences between the AC and AM genotypes may result from the adaptation of the virus to different hosts and the existence of different subgroups of the AC genotype based on regional variations (*Choe et al., 2012a*; *Grabensteiner et al., 2001*; *Ma et al., 2013b*). The AC genotype SBV strains were mainly isolated from Asian countries, and their hosts include *A. cerana* (*Zhang et al., 2001*; *Zhang, 2012*; *Reddy et al., 2016*, *2017*; *Ma et al., 2011c*; *Nguyen & Le, 2013*; *Choe et al., 2012b*; *Xia, Zhou & Wei, 2015*; *Hu et al., 2016*; *Yu, Liu & Wang, 2016*). However, the AM genotype SBV persists in the bee colony and may cause infection of *A. mellifera* (*Allen & Ball, 1996*; *Berenyi et al., 2006*; *Ellis & Munn, 2005*; *Cavigli et al., 2016*; *Tsevegmid, Neumann & Yañez, 2016*; *Desai & Currie, 2016*). The AC genotype isolated from China, infecting the Chinese honey bee *A. cerana* was named as CSBV (*Zhang, 2012*; *Qin, 2000*; *Ma et al., 2011c*, *2013b*; *Hu et al., 2016*; *Zhang et al., 2001*). CSBV and SBV are similar in structure, physiology, and biochemistry, but have different antigenicity and exhibit no cross-infection (*Yang, Zhang & Du, 1979*; *Feng, Yu & Zhang, 1999*).

Among the AC genotype SBV strains, the first genome sequence was obtained for CSBV-GD-2002 (*Zhang et al., 2001*), and subsequently other CSBV genomes, including

CSBV-FZ-2014, CSBV-JLCBS-2014, CSBV-LNQY-2008, CSBV-SXYL-2015, and CSBV-BJ-2012, were sequenced. CSBV is similar to AmSBV-UK-2000 in terms of its physiological and biochemical features; however, the viruses differ in their antigenicity and do not show cross-infection (*Cheng et al., 2011*; *Choe et al., 2012a*; *Ma et al., 2013b*). Sequence analysis indicated that CSBV was different but highly homologous to AmSBV-UK-2000. The genetic characterization and phylogenetic relationship of SBV-infected honeybees collected from different hosts and various geographic regions have recently attracted attention. Previous studies have focused on the alignment, basic structure, and composition of SBV/CSBV genomes, and host-specificity and geographic differences of SBV/CSBV (*Choe et al., 2012a*, *2012b*; *Grabensteiner et al., 2001*; *Reddy et al., 2016*, *2017*; *Ma et al., 2011c*, *2013b*; *Nguyen & Le, 2013*; *Xia, Zhou & Wei, 2015*; *Hu et al., 2016*; *Yu, Liu & Wang, 2016*; *Zhang et al., 2001*); however, it is unclear whether there is recombination between SBV and CSBV and if CSBV breaks through the species barrier, develops cross-infection, and causes disease (kills larvae) in *A. mellifera*. Using artificial infection experiments, *Gong et al. (2016)* demonstrated that CSBV is able to infect *A. mellifera*, but did not observe obvious signs of the disease, which indicated low pathogenicity.

In this study, we characterized a CSBV strain in *A. mellifera* from Shandong, China (henceforth referred to as AmCSBV-SDLY-2016 (GenBank accession No. MG733283)), and report its molecular and biological characteristics. Furthermore, we analyzed the virus recombination.

## METHODS

### Clinical sample collection and testing

In total, 359 samples (each collected sample included five larvae from a single colony) were obtained from China between 2008 and 2017. Among all 359 samples, 20 samples were from *A. mellifera* and 359 samples were from *A. cerana*. In all larvae, CSBV infection was detected by real-time quantitative RT-PCR (qRT-PCR) (*Ma et al., 2013a*). The field studies did not involve endangered or protected species, and the owner of Linyi bee farm Zhang Youcai gave permission to conduct the study on this site.

We randomly selected one positive larva for genetic characterization of VP1 from the same colony at the same time. The VP1 gene of CSBV was amplified using VP1-specific primers (F: 5′-GCGGATCCATGGATAAACCGAAGGATATAAG-3′, R: 5′-GCAAGCTTTTATTGTACGCGCGGTAAATA-3′) and sequenced.

### Phylogenetic tree construction from VP1

Using VP1 as a target gene, we constructed a phylogenetic tree from all the CSBV isolates and reference strains in GenBank (Tables 1 and 2) using the MEGA 5.0 package (*Tamura et al., 2007*) and the neighbor-joining (NJ) method (*Saitou & Nei, 1987*). The phylogenetic tree was computed using the Kimura 2-parameter method (*Kimura, 1980*). A bootstrap value of 1,000 replicates was applied to yield a robust phylogeny. We named the samples after the strain that originated in the same region and showed

**Table 1 A total of 31 CSBV strains isolated.**

| No. | Isolated strain | Region | Host | GenBank number |
|---|---|---|---|---|
| 1 | CSBV-LNQY-2008 | Qingyuan, Liaoning | *A. cerana* | HM237361 |
| 2 | CSBV-LNBX-2009 | Benxi, Liaoning | *A. cerana* | JX854438 |
| 3 | CSBV-LNSZ-2011 | Suizhong, Liaoning | *A. cerana* | JX854441 |
| 4 | CSBV-JLCC-2011 | Changchun, JiLin | *A. cerana* | JX854437 |
| 5 | CSBV-LNND-2011 | Kuandian, Liaoning | *A. cerana* | JX854439 |
| 6 | CSBV-HBQHD-2012 | Qinghuangdao, Hebei | *A. cerana* | JX854436 |
| 7 | CSBV-LNQY-2012 | Qingyuan, Liaoning | *A. cerana* | JX854440 |
| 8 | CSBV-JLCBS-2014 | Changbaishan, Jilin | *A. cerana* | KU57466 |
| 9 | CSBV-LNDD-2015 | Dandong, Liaoning | *A. cerana* | KX254334 |
| 10 | CSBV-LNJZ-2015 | Jinzhou, Liaoning | *A. cerana* | KX254336 |
| 11 | CSBV-LNQY-2015 | Qingyuan, Liaoning | *A. cerana* | KX254337 |
| 12 | CSBV-GZGY-2015 | Guiyang, Guizhou | *A. cerana* | KX254332 |
| 13 | CSBV-JXJJ-2015 | Jiujiang, Jiangxi | *A. cerana* | KX254333 |
| 14 | CSBV-SXYL-2015 | Yulin, Shanxi | *A. cerana* | KU574662 |
| 15 | CSBV-SXXA-2015 | Xian, Shanxi | *A. cerana* | KX254338 |
| 16 | CSBV-BJ-2015 | Beijing | *A. cerana* | KX254340 |
| 17 | CSBV-HB-2016 | Tangshan, Hebei | *A. cerana* | KY363865 |
| 18 | CSBV-HBQHD-2016 | Qinhuangdao, Hebei | *A. cerana* | KY363866 |
| 19 | CSBV-HNSX-2016 | Songxian, Henan | *A. cerana* | KY379055 |
| 20 | CSBV-LNHLD-2016 | Huluodao, Liaoning | *A. cerana* | KY379056 |
| 21 | CSBV-SXYC-2016 | Yuncheng, Shanxi | *A. cerana* | KY379058 |
| 22 | AcCSBV-SDLY-2016 | Linyi, Shandong | *A. cerana* | MF150113 |
| 23 | AmCSBV-SDLY-2016 | Linyi, Shandong | *A. mellifera* | MG733283.1 |
| 24 | CSBV-HBCD-2016 | Chengde, Hebei | *A. cerana* | MG979656 |
| 25 | CSBV-LNDD-2017 | Dandong, Liaoning | *A. cerana* | MG979655 |
| 26 | CSBV-SDLC-2017 | Liaocheng, Shandong | *A. cerana* | MG979657 |
| 27 | CSBV-HBCD-2017 | Chengde, Hebei | *A. cerana* | MG979658 |
| 28 | CSBV-JLHD-2017 | Huadian, Jilin | *A. cerana* | MG979659 |
| 29 | CSBV-JLTH-2017 | Tonghua, Jilin | *A. cerana* | MG979660 |
| 30 | CSBV-HNNY-2017 | Nanyan, Henan | *A. cerana* | MG979661 |
| 31 | CSBV-JXJJ-2017 | Jiujiang, Jiangxi | *A. cerana* | MG979662 |

100% homology, then submitted the data to GenBank (CSBV uniformly renamed as CSBV + first isolated geographic location + year).

## Virus purification

AmCSBV-SDLY-2016 was obtained from a natural outbreak on Linyi bee farm in Shandong, China. *A. mellifera* larvae infected by AmCSBV-SDLY-2016 were collected by the owner of Linyi bee farm. A total of 50 infected *A. mellifera* larvae displaying upward warping of the head and body surface changes, and white or sick and dead larvae in the honeycomb with the capped brood abnormally sealed, were collected, weighed, and completely homogenized in sterile water (1.5× by weight) using a mortar and pestle.

**Table 2  Reference strain and its GenBank number.**

| No. | Isolated strain | Host | GenBank number | No. | Isolated strain | Host | GenBank number |
|---|---|---|---|---|---|---|---|
| 1 | CSBV-FZ-201 | A. cerana | KM495267 | 23 | AmSBV-Kor2-2016 | A. mellifera | KP296801 |
| 2 | CSBV-GZ-2009 | A. cerana | AF251124 | 24 | AcSBV-Kor4-2016 | A. cerana | KP296803 |
| 3 | CSBV-GD-2002 | A. cerana | AF469603 | 25 | AmSBV-Kor1-2016 | A. mellifera | KP296800 |
| 4 | CSBV-BJ-2012 | A. cerana | KF960044 | 26 | AcSBV-Indk1A-2013 | A. cerana | JX270796 |
| 5 | AcSBV-V1-2014 | A. cerana | KM884990 | 27 | AcSBV-IndII-2-2013 | A. cerana | JX270795 |
| 6 | AcSBV-V2-2014 | A. cerana | KM884991 | 28 | AcSBV-IndiaS2-2013 | A. cerana | JX270799 |
| 7 | AcSBV-V3-2014 | A. cerana | KM884992 | 29 | AcSBV-K3A-2013 | A. cerana | JX270798 |
| 8 | AmSBV-V4-2014 | A. mellifera | KM884993 | 30 | AcSBV-Indiall10-2013 | A. cerana | JX194121 |
| 9 | AcSBV-V5-2014 | A. cerana | KM884994 | 31 | AcSBV-Indiall9-2013 | A. cerana | JX270800 |
| 10 | AcSBV-VHYnor-2014 | A. cerana | KJ959614 | 32 | AcSBV-IndK5B-2013 | A. cerana | JX270797 |
| 11 | AmSBV-Viet6-2014 | A. mellifera | KM884995 | 33 | AmSBV-Sweden-2017 | A. mellifera | KY273489 |
| 12 | AcSBV-VBP-2017 | A. cerana | KX668139 | 34 | AmSBV-NT-2017 | A. mellifera | KY465679 |
| 13 | AcSBV-VNam-2015 | A. cerana | KJ959613 | 35 | AmSBV-QLD-2017 | A. mellifera | KY465678 |
| 14 | AcSBV-VBG-2017 | A. cerana | KX668141 | 36 | AmSBV-WA1-2017 | A. mellifera | KY465672 |
| 15 | AmSBV-VNA-2017 | A. mellifera | KX668140 | 37 | AmSBV-WA2-2017 | A. mellifera | KY465671 |
| 16 | AcSBV-SBM2-2013 | A. cerana | KC007374 | 38 | AmSBV-TAS-2017 | A. mellifera | KY465676 |
| 17 | CSBV-CQ-2012 | A. cerana | KC285046 | 39 | AmSBV-SA-2017 | A. mellifera | KY465677 |
| 18 | CSBV-JXNC-2013 | A. cerana | KM232611 | 40 | AmSBV-VN2-2017 | A. mellifera | KY465674 |
| 19 | AmSBV-Kor19-2012 | A. mellifera | JQ390592 | 41 | AmSBV-VN1-2017 | A. mellifera | KY465675 |
| 20 | AcSBV-Kor-2011 | A. cerana | HQ322114 | 42 | AmSBV-VN3-2017 | A. mellifera | KY465673 |
| 21 | CSBV-SXnor-2012 | A. cerana | KJ000692 | 43 | AmSBV-Australia-2014 | A. mellifera | KJ629183 |
| 22 | AmSBV-Kor21-2012 | A. mellifera | JQ390591 | 44 | AmSBV-UK-2000 | A. mellifera | AF092924 |

AmCSBV-SDLY-2016 purification was performed by cesium chloride gradient centrifugation, according to the method reported by *Ma et al. (2011b*, *2011c*). The supernatant was extracted with an equal volume of 1,1,2-trichlorotrifluoroethane before the aqueous phase was layered over a discontinuous CsCl gradient (1.5 and 1.2 g/cm$^3$) and centrifuged at 270,000×$g$ for 1 h. The material at the CsCl interface was harvested and adjusted to a volume of five mL (final density 1.38 g/cm$^3$) with CsCl solution and centrifuged at 270,000×$g$ overnight. The supernatant was then successively passed through 0.45 and 0.22-µm cell filters. Healthy larvae treated by the same method were used as the negative control. CSBV was subsequently identified by reverse-transcription polymerase chain reaction (RT-PCR) to exclude black queen cell virus (BQCV), acute bee paralysis virus (ABPV), chronic bee paralysis virus (CBPV), deformed wing virus (DWV), Kashmir bee virus (KBV), and Israeli acute paralysis virus (IAPV), following the method reported by *Yu, Liu & Wang (2016)*. CSBV virus samples free of other viruses were stored at −80 °C until further use.

## Electron microscopy and sodium dodecyl sulfate polyacrylamide gel electrophoresis for virus identification

As previously described (*Ma et al., 2011b*, *2011c*), 100 µL of the purified viral suspension was directly pelleted onto carbon-coated Formvar copper grids by ultracentrifugation

(15 min at 82,000×$g$) using a Beckman Airfuge. The grids were negatively stained with 2% sodium phosphotungstate at pH 6.8 for 90 s and observed using a Philips CM10 transmission electron microscope.

Structural proteins were separated by sodium dodecyl sulfate polyacrylamide gel electrophoresis (SDS-PAGE) with 5% stacking and 12% separating gels using standard protocols.

## Agar gel immunodiffusion assay

Briefly, one g of agarose and eight g of sodium chloride (NaCl) were added to 100 mL phosphate buffer (0.01M, pH 7.2), shaken well, and microwaved for 2 min to prepare an agar solution. The solution was slightly cooled, poured into Petri dishes (90 mm in diameter; 20–22 mL of agar per plate), and allowed to solidify. Seven wells were made in the agar plates to identify AmCSBV-SDLY-2016. The central hole was loaded with antisera against the CSBV-JLCBS-2014 strain; the surrounding wells 1 and 2 were loaded with purified AmCSBV-SDLY-2016; wells 3 and 4 were loaded with a known positive control (purified CSBV-JLCBS-2014); and wells 5 and 6 were loaded with treated healthy larvae as the negative control. The plates were then placed in a closed, moist container and incubated at 37 °C in a humidified chamber for 24 or 48 h. Precipitation was visible after 24 h but became more distinct after 48 h of incubation.

## Analysis of pathogenicity

As described previously (*Ma et al., 2013a*), PCR probes were used to measure the copy numbers of AmCSBV-SDLY-2016. The purified AmCSBV-SDLY-2016 was 10-fold serially diluted.

As described previously (*Hu et al., 2016*), two-day-old *A. cerana* larvae from a single mated queen honeybee in a healthy apiary in Jinzhou, Liaoning Province, were orally inoculated with CSBV. In total, 120 larvae were selected and randomly distributed into six groups ($n = 20$ larvae per group). Each group was inoculated with AmCSBV-SDLY-2016 ($1.25 \times 10^4$, $1.25 \times 10^5$, $1.25 \times 10^6$, $1.25 \times 10^7$, and $1.25 \times 10^8$ copies/larva in groups 1–6, respectively). The assay was performed in triplicate. The virus inoculation assay was performed twice after the end of the first experiment.

Each larva was fed 20 µL of a virus suspension mixed with an equal amount of basic larval diet (BLD, *Liu & Zeng, 2010*) consisting of 50% royal jelly, 37% sterile water, 6% glucose, 6% fructose, and 1% yeast extract, at 95% relative humidity and 34 °C. The virus-free control was fed 20 µL of sterile water with an equal amount of BLD. BLD was used subsequently for daily feeding. The clinical signs in each group of larvae were examined and recorded every day until larvae death. All larvae that were orally inoculated in this study were analyzed by RT-PCR for BQCV (*Grabensteiner et al., 2007*), ABPV (*Grabensteiner et al., 2007*), CBPV (*Tentcheva et al., 2004*), DWV (*Tentcheva et al., 2004*), KBV (*Blanchard et al., 2014*), IAPV (*De Miranda, Cordoni & Budge, 2010*), and qRT-PCR for CSBV (*Ma et al., 2013a*).

The statistical analysis of larval mortality was performed between groups and between three replicates of each group using SPSS 22.0.

**Table 3 Synthetic oligonucleotides for amplification of the CSBV genome.**

| Primers | Sequence (5′ to 3′) | Nucleotide position |
|---|---|---|
| S1 F | GAAATAAGAATACGAATCGT | 1–20 |
| S1 R | TAAACAAATCGGTATAAGAGTCC | 379–401 |
| S2 F | GACCCGTTTTCTTGTGAGTTTTAG | 41–64 |
| S2 R | GTGTAGCGTCCCCCTGAATAGAT | 611–633 |
| S3 F | CGTAATTGCGGAGTGGAAAGAT | 273–294 |
| S3 R | CCCCTAAATTGTTGCGTTGGTT | 739–760 |
| S4 F | TATTCAGGGGGACGCTACAC | 614–633 |
| S4 R | TATTCCATCGGGGTTATTTG | 1,713–1,732 |
| S5 F | GGAGACGCGCATGGTAAAGA | 1,644–1,663 |
| S5 R | GCGCGGTAAATAAACACTCG | 2,365–2,384 |
| S6 F | ATGGGGGTAAGGGACAATCTG | 2,290–2,310 |
| S6 R | TGCTCTAACCTCGCATCAAC | 3,423–3,442 |
| S7 F | TTACGGGAGCAGCACAACA | 3,391–3,409 |
| S7 R | ATTTCCGATTTACCGATACC | 4,287–4,306 |
| S8 F | CGGTGCGTTATGAACCTTTT | 4,243–4,262 |
| S8 R | AATGCGTAGATTGAGGTGCC | 5,333–5,352 |
| S9 F | TACTATCCGCCCCCTAAGC | 3,195–3,213 |
| S9 R | GTGCCCCATCGTTCAAAA | 5,432–5,449 |
| S10 F | GCGCAACTGGCACCTCAAT | 5,325–5,343 |
| S10 R | TTCCAAATATACTTCCCACTGC | 6,249–6,270 |
| S11 F | GCTGGGCCTTCTTATCTGGTG | 3,873–3,893 |
| S11 R | TACGGGTCCTCTGCAATGTTCT | 6,363–6,384 |
| S12 F | GTGACGGCAGTGGGAAGTAT | 6,262–6,243 |
| S12 R | GCAGCCTCCTCAGGTGTTAGT | 7,454–7,474 |
| S13 F | TTTGGTAGCGGGGTGTAAG | 7,322–7,340 |
| S13 R | CATTGCGTGGTATCATT | 8,501–8,517 |

## AmCSBV-SDLY-2016 genome sequencing

The primers used in this study (Table 3) were designed based on the nucleotide sequences of CSBV-JLCBS-2014, AmSBV-UK-2000, and CSBV-GD-2002. The full-length AmCSBV-SDLY-2016 genome was determined by Sanger sequencing of the RT-PCR fragments and 3′ rapid amplification of cDNA ends (RACE) (Clontech-Takara, Mountain View, CA, USA), according to the method described by *Ma et al. (2011c)*. The PCR amplification product was cloned into the pMD-18-T vector (Takara Biotechnology Co. Ltd., Dalian, China). The plasmids were then used to transform *Escherichia coli* DH5α cells (Takara Biotechnology Co. Ltd., Dalian, China). For each RT-PCR fragment, five clones were randomly chosen and sequenced by Sangon Biotech Co. Ltd. (Shanghai, China). If all five clones were identical, the sequence was considered correct. The correct nucleotide sequences from all the fragments were assembled to build a continuous complete sequence using the DNASTAR software, and the VP1 gene sequence was analyzed as described above.

### Phylogenetic tree construction from AmCSBV-SDLY-2016 and SBV/CSBV genome sequences

Nucleotide sequences of the amplified RT-PCR fragments were assembled to generate the entire genome of AmCSBV-SDLY-2016 using the DNASTAR program. Multiple nucleotide and deduced amino acid sequence alignments identified sequence variations between clones covering the same SBV genome regions using ClustalW in the MegAlign program (DNAStar Inc., Madison, WI, USA) and the published SBV/CSBV sequences. A phylogenetic tree was constructed from the nucleotide sequences of the coding regions of 43 previously reported SBV/CSBV strains from various countries and the AmCSBV-SDLY-2016 isolated in this study. The phylogenetic tree constructed using the MEGA 6.0 package (*Tamura et al., 2007*) and NJ method (*Saitou & Nei, 1987*) was computed using the Kimura 2 parameter method (*Kimura, 1980*). The phylogenetic tree was bootstrapped 1,000 times and bootstrap values placed on each branch.

### Similarity and virus recombination analysis

Based on the results of the phylogenetic analysis, the isolated CSBV/SBV strains were divided into nine groups as follows: A (CSBV-FZ-2014), B (AmCSBV-SDLY-2016), C (CSBV-SXnor-2012), D (CSBV-SXYL-2015), E (CSBV-BJ-2012), F (CSBV-LNQY-2008), G (CSBV-JLCBS-2014), H (CSBV-GD-2002), and I (AmSBV-UK-2000) for similarity analysis and virus recombination analysis. The Recombination Detection Program (RDP) values were computed using the RDP 4.0 software for the preliminary screening of the major parent and minor parent sequences. Similarity plots and BootScan were computed using Simplot software (*Lole et al., 1999*) with the following parameters: a window of 200 base pairs (bp; step: 20 bp), with gap-stripping and Kimura (2-parameter) correction, using AmCSBV-SDLY-2016, CSBV-JLCBS-2014, CSBV-GD-2002, and AmSBV-UK-2000 for Find Sites, and AmCSBV-SDLY-2016 and CSBV-JLCBS-2014 as the query sequences.

## RESULTS

### Characteristics of the VP1 gene

Of the 359 samples tested, 326 were found to be positive by strand-specific qPCR, 18 of which originated from *A. cerana*, whose infection rate was 90.0% (18/20), and the 95% confidence intervals for infection rate were 69–99%, with 308 positive samples originating from *A. mellifera* whose infection rate was 90.9% (308/339), and the 95% confidence intervals for the infection rate were 85.90–95.81%. Positive samples originating in the same region that showed 100% homology were defined as a strain. A total of 31 CSBV strains were isolated according to the aforementioned virus-naming scheme.

The phylogenetic tree of VP1 revealed two clusters (Fig. 1). One was related to SBV strains that originated from *A. cerana* (AC genotype), the other was related to the SBV strains that originated from *A. mellifera* (AM genotype). The phylogenetic tree also showed that AmCSBV-SDLY-2016 originating from *A. mellifera* belonged to the clade containing the CSBV and other Asian strains. To simplify the naming of the strain, we used AmSBV and AmCSBV to represent the SBVs/CSBV strains isolated from

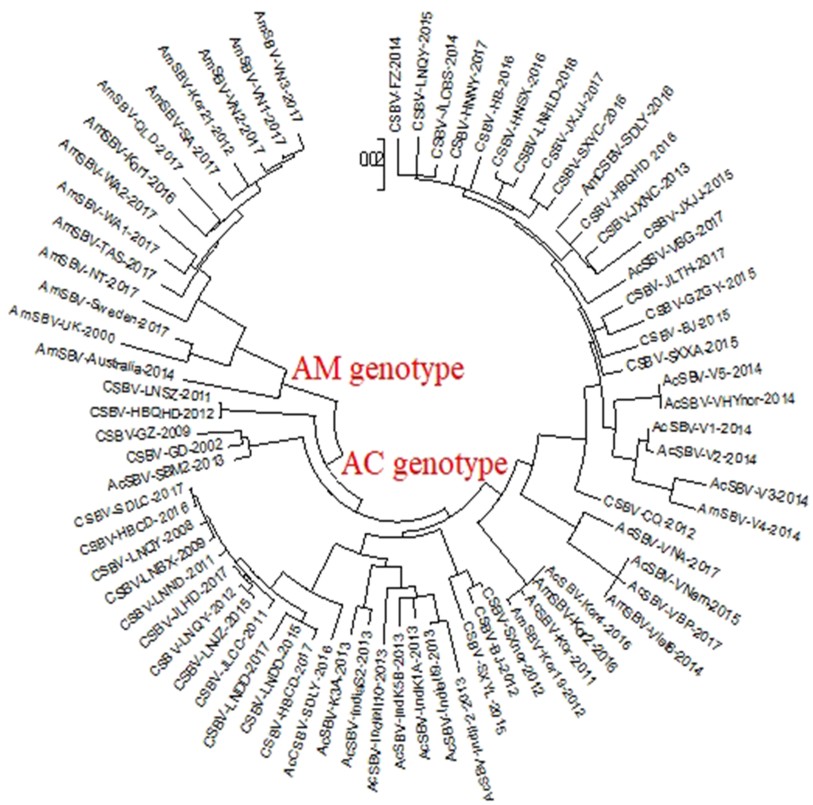

**Figure 1 Phylogenetic analysis of the VP1 gene sequences obtained from all the CSBV isolates and reference strains in GenBank.** The tree reveals two clusters, the AC and AM genotypes. Note: To simplify the naming of the strain, we used AmSBV and AmCSBV to represent the SBVs/CSBV strains isolated from *A. mellifera*; all the others represent the SBVs/CSBV strains that were isolated from *A. cerana*. The corresponding host and gene numbers are listed in Tables 1 and 2.

*A. mellifera*; all the others represent the SBVs/CSBV strains that were isolated from *A. cerana*. The corresponding host and gene numbers are listed in Tables 1 and 2.

## AmCSBV-SDLY-2016 identified by electron microscopy and SDS-PAGE

Electron microscopy showed large amounts of typical CSBV particles in the preparations from virus-infected larvae; CSBV particles were icosahedrons and had an approximate diameter of 26 nm (Fig. 2). No virus particles were observed in the control preparations from healthy larvae. The four main structural proteins of CSBV were separated by SDS-PAGE (Fig. 3). The molecular weights of the four proteins were about 44.2, 37.8, 31.5, and 30.5 kDa, respectively (*Feng et al., 1998*; *Ma et al., 2011a*).

## AmCSBV-SDLY-2016 identified by AGID assay

Agar gel immunodiffusion (AGID) assay revealed the distinct CSBV-specific lines of precipitin observed between the wells containing AmCSBV-SDLY-2016 and the antisera against CSBV-JLCBS-2014, as well as the positive control wells (Fig. 4). No precipitin lines were observed in the negative control wells.

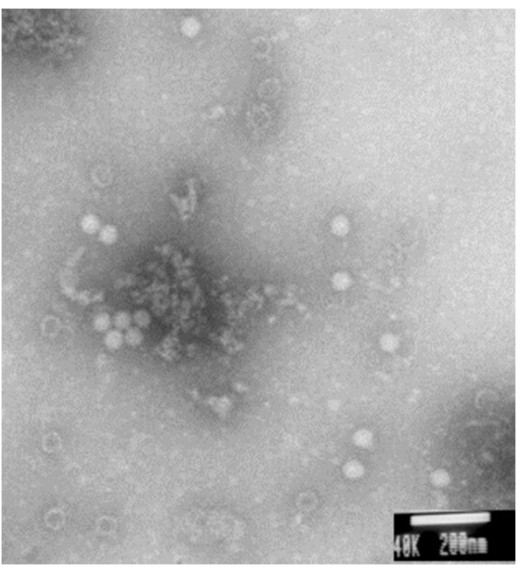

**Figure 2 CSBV particles were observed with electron microscopy.** Virus particles of approximately 26-nm diameter were observed in virus preparations from the infected larvae by electron microscopy.

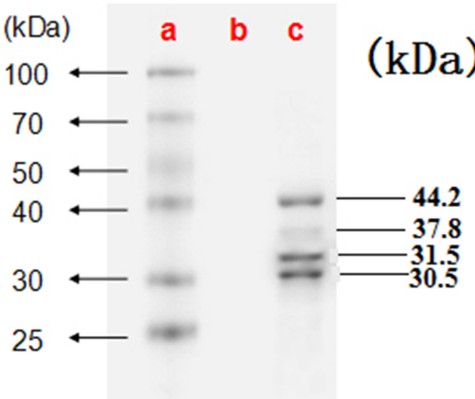

**Figure 3 The four proteins of CSBV were detected by SDS-PAGE.** The proteins were resolved on 12% SDS-polyacrylamide gels according to standard protocols. (A) Representative protein markers. (B) Virus-free control. (C) AmCSBV-SDLY.               

## Pathogenicity

After oral inoculation with AmCSBV-SDLY-2016 (groups 1–5), all larvae were analyzed by RT-PCR and qRT-PCR. All of the other honeybee viruses were undetectable, whereas CSBVs were detectable. All the larvae in the virus-free control (group 6) lacked common honeybee viruses.

Three repeated experiments showed that upon inoculation with $1.25 \times 10^4$, $1.25 \times 10^5$, or $1.25 \times 10^6$ copies of AmCSBV-SDLY-2016, larvae mortality rates were 35–45%, 65–75%, and 80–90%, whose 95% confidence intervals for the proportion dead pupae were 19–64%, 46–88%, and 62–97%, respectively (Table 4). In contrast, mortality of the larvae

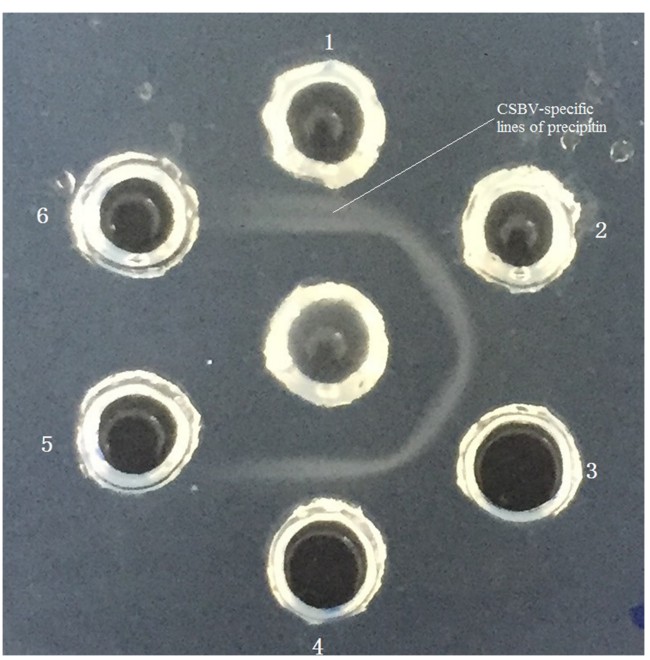

**Figure 4 Identification of AmCSBV-SDLY by AGID assay.** Antisera CSBV (center well), AmCSBV-SDLY (wells 1 and 2), known positive control CSBV-JL2014-KU574661.1 (wells 3 and 4), and negative control (wells 5 and 6).

**Table 4 Experimental results of larval challenge.**

| Group | The number of dead larvae at different time points (a/b/c) | Mortality (%) | 95% CI (%) |
|---|---|---|---|
| 1 | 7/8/9 | 35–45 | [19–64] |
| 2 | 13/15/14 | 65–75 | [46–88] |
| 3 | 17/16/18 | 80–90 | [62–97] |
| 4 | 20/20/20 | 100 | [83–100] |
| 5 | 20/20/20 | 100 | [83–100] |
| 6 | 4/4/5 | 20–25 | [6–44] |

**Note:**
a, b and c represent the number of dead larvae from three repeated experiments.

was 100% in the two groups, where each larva was sequentially inoculated with $1.25 \times 10^7$ and $1.25 \times 10^8$ copies of AmCSBV-SDLY-2016, whose 95% confidence intervals for the proportion dead pupae were 83–100%. In the virus-free control groups, the mortality rates of the larvae were 20–25%, with 95% confidence intervals for the proportion dead pupae of 6–44%.

Statistical analysis showed that there were no significant differences between the larval mortality from three repeated experiments in each group, but there were highly significant differences in larval mortality between each group, except that there was no difference in the number of larval mortality between the two groups, where each larva was sequentially inoculated with $1.25 \times 10^7$ and $1.25 \times 10^8$ copies of AmCSBV-SDLY-2016.
## Nucleotide sequence

The nucleotide sequence of the AmCSBV-SDLY-2016 genome was 8,794-bp long, and the percentages of A, U, G, and C were 29.95%, 29.22%, 24.30%, and 16.52%, respectively, similar to those reported for other CSBV/SBV strains encoding one large ORF. Two AUG codons were located at positions 189 and 429 of the AmCSBV-SDLY-2016 genome; however, AUG 189 is likely the translation initiation site, as unlike AUG 429, it was observed in a sequence (AUUAUGG) identical to that of many invertebrate initiation codons (ANNAUGG). The size of the CSBV 3′ untranslated region (UTR) (77 nucleotides) was similar to that from other picornaviruses (40–126 nucleotides). The 5′ UTR was generated from AmCSBV-SDLY-2016 by RT-PCR with a primer designed using the CSBV-JLCBS-2014 sequence, indicating that the CSBV 5′ UTR was similar to that of CSBV-JLCBS-2014 (188 nucleotides). Multiple sequence comparisons revealed a sequence homology of 92.4– 97.1% among all CSBV isolates and a similarity of 94.5–97.7% in the deduced amino acid sequences. AmCSBV-SDLY-2016 was least similar (89.5–90.4% identity) to other SBVs but showed maximum similarity with CSBV-FZ-2014 (97.1% and 97.7% homology of sequence and deduced amino acid sequences, respectively). The results of VP1 sequencing were completely consistent, so we could rule out the possibility that several CSBV isolates could co-exist in the samples originating from *A. mellifera*.

## Protein sequence

The deduced amino acid sequences of AmCSBV-SDLY-2016 genomes and previously reported SBV/CSBV strains were aligned and compared. Results revealed that the structural and non-structural proteins were located at the 5′ and 3′ ends, respectively. Multiple sequence alignment showed that AmCSBV-SDLY-2016 had 17 and one amino acid deletions (positions 711–729, which are in the region of structural protein VP1, and 2,128, which is in the region of nonstructural protein) as compared with CSBV-GD-2002, and three and 13 amino acid deletions (positions 711–713 and 715–728, which are in the region of structural protein, respectively) as compared to AmSBV-UK-2000 (Fig. 5). However, AmCSBV-SDLY-2016 was similar to the CSBV-JLCBS-2014 strain that infects *A. cerana*.

The amino acid sequence at the C-terminal region of the AmCSBV-SDLY-2016 polyprotein was similar to that for helicase, protease, and RNA-dependent RNA polymerase (RdRp) domains of the previously reported SBV/CSBV strains. The highly conserved amino acid motifs GPAGIGKS, QPVVVYDD, and KKIRGNPLIVILLCNH, corresponding to the helicase domains A, B, and C (Supplemental Information 1), respectively, were located between the amino acid positions 1,353 and 1,473; however, the C domain containing KKIRGNPLIVILLCNH appeared to be the most conserved, unlike mammalian picornaviruses. The conserved cysteine protease motif $^{2249}$GXCG$^{2252}$ (GACG) and the putative substrate-binding residue $^{2266}$GxHxxG$^{2271}$ (GMHFAG) were detected in the 3C protease domains spanning amino acids 2,141–2,272 (Supplemental Information 2). In addition, eight conserved domains identified in RdRp were also detected between amino acid positions 2,444 and 2,813 in AmCSBV-SDLY-2016 (Supplemental Information 3).

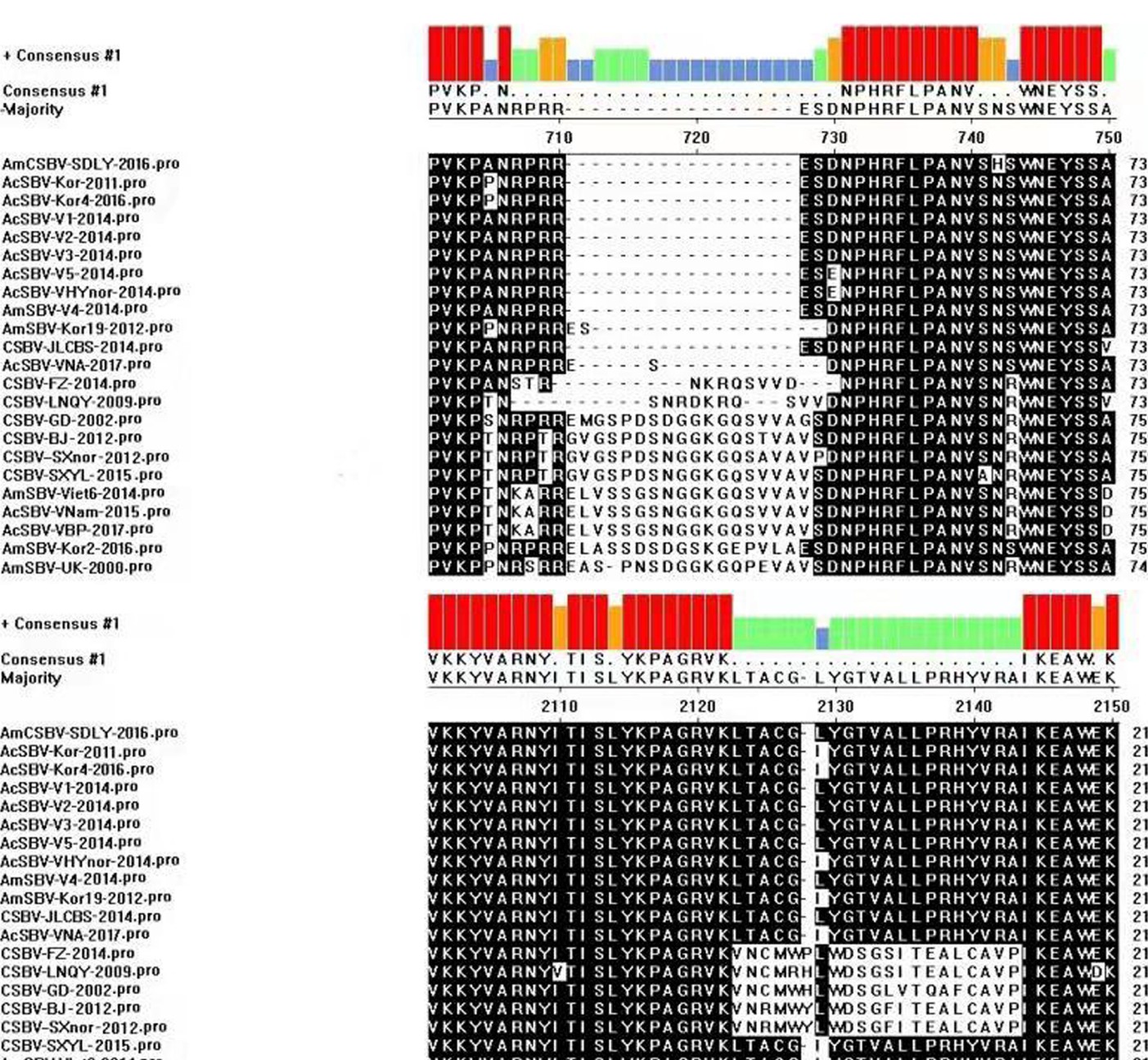

**Figure 5 Protein domain alignments.** AmCSBV-SDLY-2016 had 17 and one amino acid deletions (positions 711–729 and 2,128, respectively). However, AmCSBV-SDLY-2016 had three and 13 amino acid deletions (positions 711–713 and 715–728, respectively) as compared with AmSBV-UK-2000.

## Phylogenetics

To assess the genetic relationship among the SBV/CSBV strains, a phylogenetic tree based on the nucleotide sequence of the SBV/CSBV-coding region was constructed by the NJ method for AmCSBV-SDLY-2016 and all SBV/CSBV reference strains in GenBank.

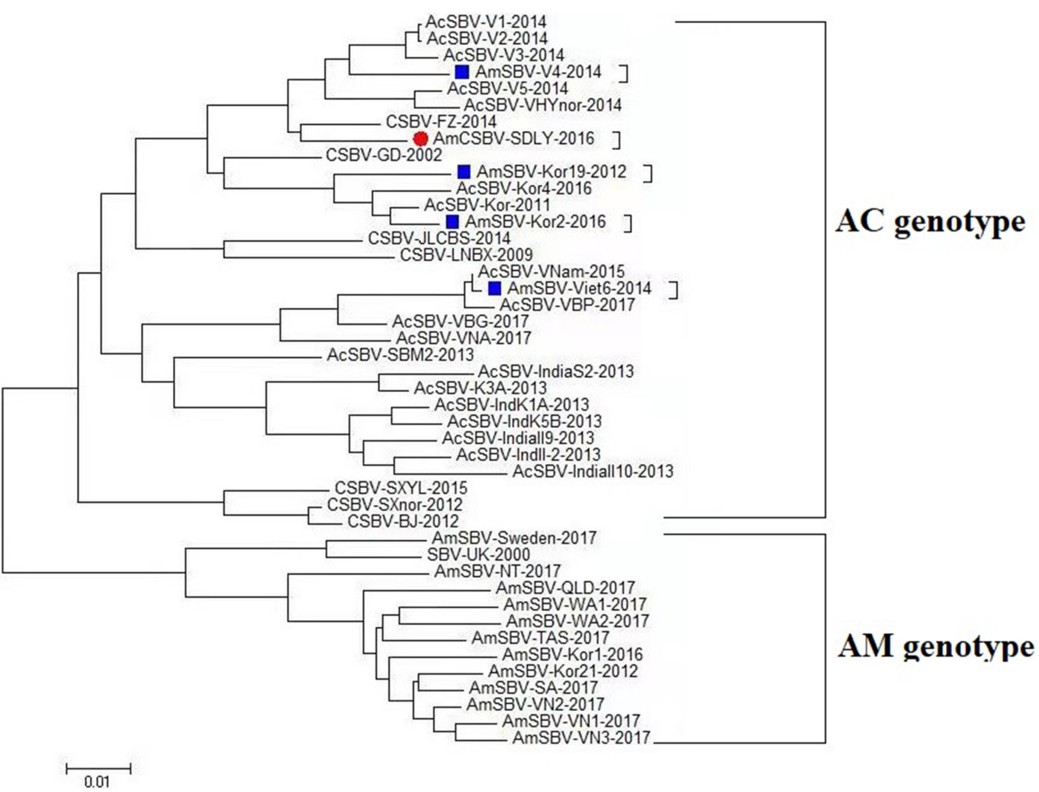

**Figure 6 Phylogenetic analysis of all nucleotide sequences obtained from various countries, including China, Korea, Vietnam, India, Australia, and the United Kingdom.** Numbers on the nodes indicate clade credibility values. Previously reported sequence names are presented in the following format: strains-GenBank accession numbers. The tree reveals two clusters, the AC and AM genotypes. The bar represents a genetic distance of 0.01. CSBV is marked with a red box, the blue square represents that the AC genotype SBVs strains mainly were isolated from *A. mellifera*.

The tree was similar to the phylogenetic tree of VP1, which revealed two clusters, one related to SBV strains that originated from *A. cerana* (AC genotype), and the other related to the SBV strains that originated from *A. mellifera* (AM genotype) (Fig. 6). The AM cluster was distinctly separate from the AC lineages, as supported by a clade credibility value of 100.0. The AC genotype was further subdivided into several subtypes according to their countries of origin and host species.

The phylogenetic tree also showed that AmCSBV-SDLY-2016 belonged to the clade containing the CSBV and other Asian strains. These results indicate that the AmCSBV-SDLY-2016 strain showed close genetic relationships with CSBV strains, specifically CSBV-FZ-2014.

## Similarity profile

To identify differences in the full-length sequences and various genomic regions from CSBV strains AmSBV-UK-2000 and AmCSBV-SDLYA-2016, the complete coding regions were plotted using SimPlot, with AmCSBV-SDLY-2016 as the query sequence (Fig. 7). AmCSBV-SDLY-2016 from *A. mellifera* was more similar to the CSBV strains isolated
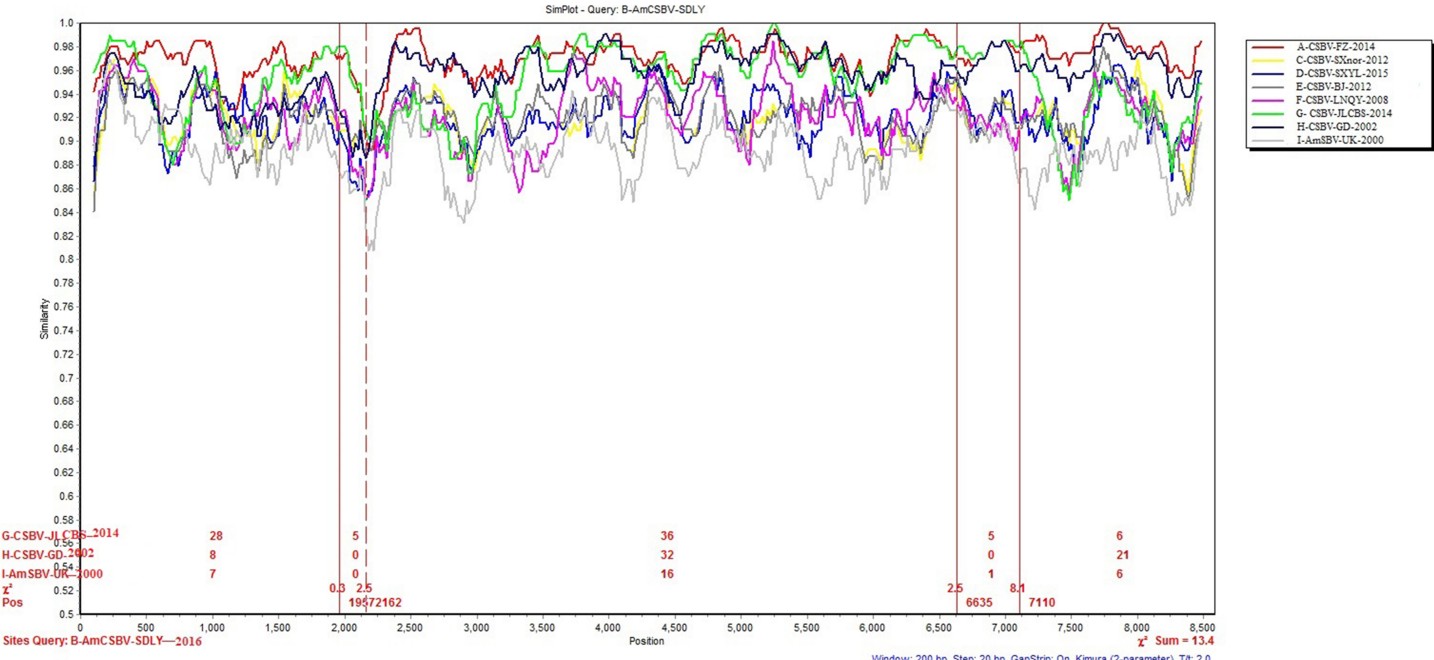

**Figure 7 SimPlot graphs comparing AmCSBV-SDLY-2016 with other CSBVs and AmSBV-UK-2000.** Each colored line indicates a group of SBV/CSBV strains (for interpreting the color key in this figure legend, please refer to an online version of the article).

**Table 5 Recombinants detected with RDP.**

| Recombinant | Major parent | Minor parent | Breakpoint | Av. *P*-val |
|---|---|---|---|---|
| CSBV-JLCBS-2014 | CSBV-FZ-2014 | CSBV-LNQY-2008 | 520–1,277 | $1.048 \times 10^{-18}$ |
| | AmCSBV-SDLY-2016 | CSBV-LNQY-2008 | 2,212–3,247 | $8.240 \times 10^{-18}$ |
| | CSBV-FZ-2014 | CSBV-LNQY-2008 | 7,202–8,336 | $7.771 \times 10^{-17}$ |

than AmSBV-UK-2000. The CSBV strains isolated formed a relatively independent separation group; there was an obvious deviation compared to AmSBV-UK-2000, with maximum deviation (19.3%) occurring at 2,181 and 2,221 bp. A high degree of consistency was observed between the genomes of AmCSBV-SDLY-2016 and isolated CSBV strains, with more than 85% similarity between the complete coding regions. Among all the isolated CSBV strains, only recombinant CSBV-JLCBS-2014 was detected by RDP (Table 5) that exhibited three recombinant fragments as follows: one obtained from AmCSBV-SDLY-2016 and two derived from CSBV-FZ-2014. The minor parent was CSBV-LNQY-2008. BootScan analysis showed that the cross-recombination signals of AmCSBV-SDLY-2016 and CSBV-FZ-2014 were detected at positions 1,957–2,162 and 6,635–7,110 using AmCSBV-SDLY-2016 and CSBV-JLCBS-2014 as the query sequence, respectively (Fig. 8).

## DISCUSSION

Chinese SBV was first described in Guangdong in 1972; it later disseminated to the rest of China and Southeast Asian countries, causing a lethal disease in individual bees or the

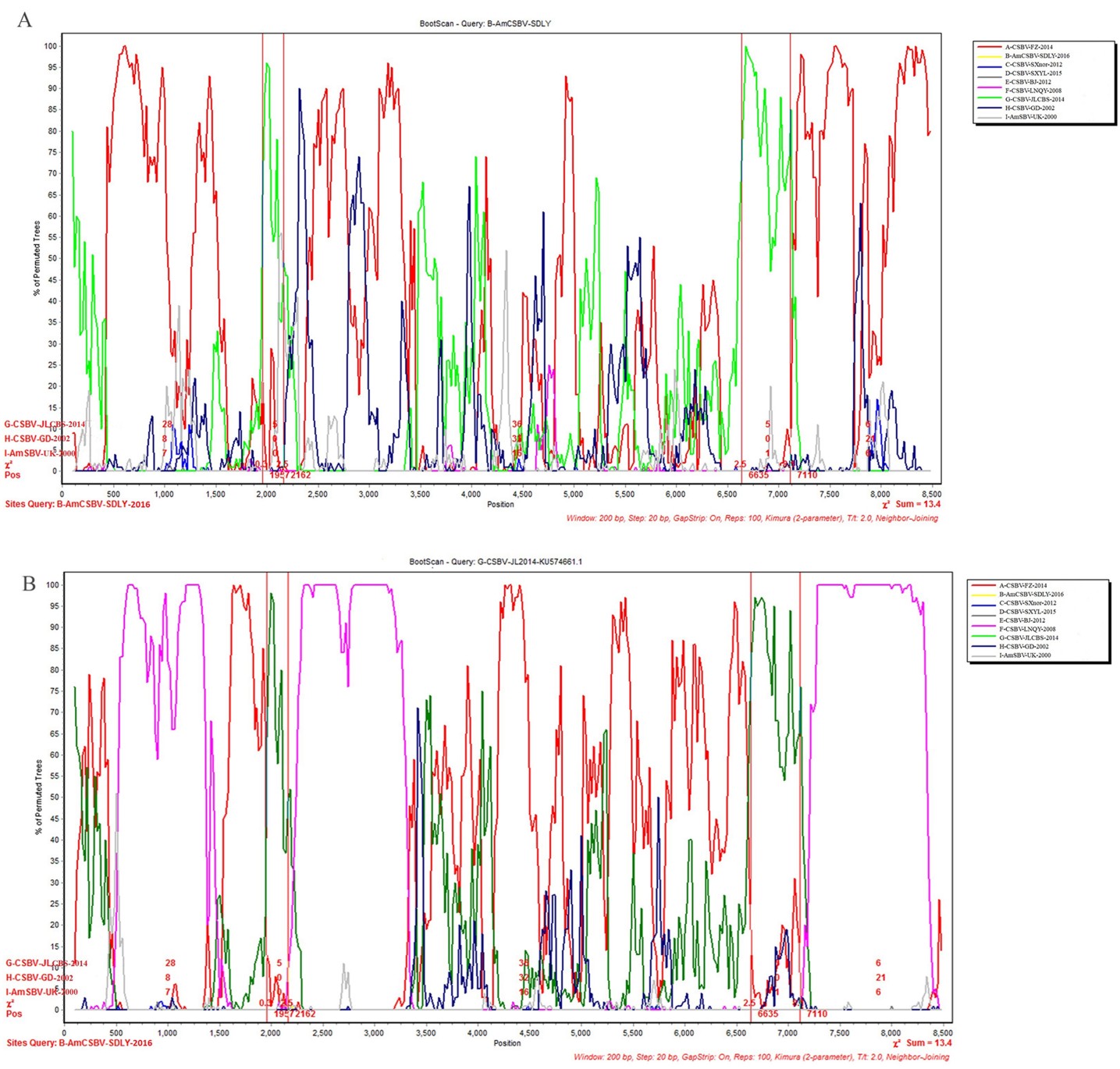

**Figure 8 BootScan analysis of the recombinant sequence based on the comparison between the complete coding regions of AmCSBV-SDLY-2016 and other CSBVs and AmSBV-UK-2000.** (A) BootScan was conducted with the strain AmCSBV-SDLY-2016 as the query sequence. (B) BootScan was conducted with the strain CSBV-JLCBS-2014 as the query sequence. Note: The cross recombination position is labeled by red lines.

collapse of entire colonies, and was termed as "bee cancer" by beekeepers (*Ma et al., 2010*). More than 200,000 colonies (more than half of the total number of local colonies) were destroyed in Guangxi and Guangdong provinces from 1972 to 1994 (*Qin, 2000*). It re-emerged and caused the collapse of entire colonies in Liaoning China in 2008

(*Zhang, 2012*; *Ma et al., 2010*). Although CSBV and SBV are similar in structure, physiology, and biochemistry, these viruses differ in antigenicity and cross-infection. SBV strains usually show host specificity in the structural polyprotein regions of their genomes (*Cheng et al., 2011*; *Choe et al., 2012a*; *Ma et al., 2013b*). CSBV, which has been divided into different subtypes according to various geographic regions and host specificity (*Ma et al., 2013b*), contains three structural proteins, namely, VP1, VP2, VP3, and an unknown protein with a molecular weight of 44.2 kDa (*Feng et al., 1998*; *Ma et al., 2011a*). SBV contains three structural proteins (VP1, VP2, and VP3). The proteins VP1, VP2, and VP3, as well as the unknown 44.2 kDa protein of CSBV, were detected by SDS-PAGE on CSBV strains (termed as AmCSBV-SDLY-2016) that infect *A. mellifera*, and distinct CSBV-specific lines of precipitation were observed between AmCSBV-SDLY-2016 and the antisera against CSBV-JLCBS-2014 in the AGID assay. Thus, AmCSBV-SDLY-2016 belongs to CSBV. As expected, the phylogenetic and similarity analyses showed that the AmCSBV-SDLY-2016 genome and the specific VP1 gene isolated from *A. mellifera* were highly similar to the genomes and specific VP1 gene of CSBV strains derived from *A. cerana*. Similar to AmCSBV-SDLY-2016, two Vietnamese SBVs strains (AmSBV-Vt-4-2014 and AmSBV-Viet6-2014) and two Korean strains (AmSBV-Kor19-2012 and AmSBV-Kor2-2016) isolated from *A. mellifera* also belong to the AC type (Fig. 6), consistent with previous reports (*Reddy et al., 2017*; *Choe et al., 2012b*). Although no reports are available to determine if this strain caused *A. mellifera* larval deaths, the results of the present study show that the long-term existence and widespread prevalence of the AC type SBV in nature may lead to exchange of viruses among host populations. Furthermore, phylogenetic and similarity analyses showed that the virus strains from the same continent or country have higher levels of similarity and genetic clustering.

The phylogenetic tree based on the nucleotide sequences of the SBV/CSBV-coding region revealed the genetic relationship between different virus isolates obtained from various geographic areas and hosts. Two distinct branches were formed according to bee species, and a higher level of similarity was observed between viral genomes from the same country, continent, or neighboring countries. Furthermore, CSBV genome sequences were closely related to those from Vietnam, Korea, and India, all of which formed a neighboring subcluster. The phylogenetic trees based on the SBV/CSBV-coding region and VP1 nucleotide sequences were identical (*Ma et al., 2013b*), indicating that SBV/CSBV strains may be divided into different groups, and that VP1 may replace the complete genomic sequence of SBVs and CSBVs and be used as a specific target for studying genotyping, genetic evolution, and molecular epidemiology of SBV/CSBV strains in the future.

Direct sequencing of virus genome amplicons and comparison with known nucleotide sequences or deduced protein sequences of virus isolates obtained from various geographic areas and hosts may reveal the genome organization and molecular basis of pathogenicity of the newly isolated strains. Analysis of the deduced amino acid sequences showed that AmCSBV-SDLY-2016, like CSBV, comprised conserved motifs within the helicase, protease, and RdRp domains. Furthermore, AmCSBV-SDLY-2016 from CSBVs

was genetically similar to the previously reported strains CSBV-FZ-2014 and CSBV-JLCBS-2014. We observed nucleotide transitions and insertions in the entire genome sequences, thereby reflecting the divergence of these genomes. In comparison to the CSBV strains that were isolated prior to 2014 (CSBV-GD-2002, CSBV-LNQY-2008, CSBV-FZ-2014, CSBV-SXnor-2012, and CSBV-BJ-2012), AmCSBV-SDLY-2016, CSBV-JLCBS-2014, CSBV-SXYL-2015, and AmSBV-UK-2000 strains carried a deletion mutation at the amino acid residue 2,128, and AmCSBV-SDLY-2016 and CSBV-SXYL-2015 differed at positions 711–729. We speculated that if AmCSBV-SDLY-2016 could cross the species barrier and cause death in *A. mellifera* larvae, then CSBV-JLCBS-2014 could probably cross the species barrier and kill *A. mellifera* larvae because of the similarity at positions 711–729 and 2,128. In addition, we demonstrated that CSBV-JLCBS-2014 was a recombinant virus, and the sequence of its parent strain was derived from AmCSBV-SDLY-2016, CSBV-FZ-2014, and CSBV-LNQY-2008 as determined by RDP and BootScan. However, AmCSBV-SDLY-2016 was isolated after CSBV-JLCBS-2014, and the recombination signals obtained from CSBV-FZ-2014, CSBV-JLCBS-2014, and CSBV-GD-2002 were detected by BootScan (Fig. 7). Hence, we speculate that AmCSBV-SDLY-2016 may be a recombinant virus strain and its parent strain was derived from CSBV-FZ-2014, CSBV-JLCBS-2014, and CSBV-GD-2002.

## CONCLUSIONS

In summary, the strain AmCSBV-SDLY-2016 of CSBV is genetically similar to the previously reported strains of CSBV that infect *A. cerana*; however, AmCSBV-SDLY-2016 may cross the species barrier, and infect and cause death in *A. mellifera* larvae. This observation shows that under natural selection and immune pressure, CSBV/SBV strains may be exchanged between hosts, indicative of the independent evolution of these viruses. Future studies will evaluate genetic variations and population structures of CSBV/SBV strains infecting *A. mellifera* and *A. cerana* in China and other countries.

### Funding

This work was supported by grants from the National Science Foundation of China (grant No. 31772760) and the Liaoning Natural Science Foundation (grant No. 20180550289). The funders had no role in study design, data collection and analysis, decision to publish, or preparation of the manuscript.

### Grant Disclosures

The following grant information was disclosed by the authors:
National Science Foundation of China: 31772760.
Liaoning Natural Science Foundation: 20180550289.

### Competing Interests

The authors declare that they have no competing interests.

## Author Contributions

- Ming Li conceived and designed the experiments, performed the experiments, analyzed the data, authored or reviewed drafts of the paper, approved the final draft.
- Dongliang Fei performed the experiments, analyzed the data, authored or reviewed drafts of the paper, approved the final draft.
- Li Sun performed the experiments, contributed reagents/materials/analysis tools, prepared figures and/or tables, authored or reviewed drafts of the paper, approved the final draft.
- Mingxiao Ma conceived and designed the experiments, authored or reviewed drafts of the paper, approved the final draft.

## Field Study Permissions

The following information was supplied relating to field study approvals (i.e., approving body and any reference numbers):

The field studies did not involve endangered or protected species, and the owner of the Linyi bee farm, Zhang Youcai, gave permission to conduct the study on this site.

## Data Availability

The raw data is available at GenBank: MG733283.

## Supplemental Information

Supplemental information for this article can be found online at http://dx.doi.org/10.7717/peerj.8003#supplemental-information.

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
