# Peer review of "Genetic and phylogenetic analysis of Chinese sacbrood virus isolates from Apis mellifera"

_PeerJ, doi:10.7717/peerj.8003_

## Round 0.1 · original submission · Major Revisions

Please pay special attention to the comments regarding the need of more details about the biological features of the isolates, and also about the materials and methods of the genome sequencing.,

·

Basic reporting

This paper reports sequence of a strain of Chinese Sacbrood virus (CSBV strain named AmCSBV-SDLY) which was isolated from Apis mellifera. This isolate showed highest similarity to the previously reported CSBV isolates from Apis cerana (with highest similarity with the GenBank entry KM495267 reaching 97% nt / 98% amino acid identity).

The main drawback of the MS is the lack of biological data on this CSBV strain. The authors did not provide data on how widespread AmCSBV-SDLY is in Apis mellifera in China. This could be addressed by devising strand specific qPCR for AmCSBV-SDLY.

Also, additional information have to be provided to prove that the published sequence (which was assembled using several RT-PCR fragments) indeed corresponds to a single RNA genome. It cannot be ruled out that several CSBV isolates could co-exist in the sampled honeybees.

Experimental design

The sequence was assembled using a series of RT-PCR fragments (n=8, according the primers shown in Table 1) and a 3'RACE products which together covered entire genome. The authors should specify how many plasmid clones were sequenced for each RT-PCR fragment and if any nt variation was observed between the clones covering the same region. It is also not clear yet if there were variation in the overlapping sections of short RT-PCR fragments used.


Other comments ( also included in Annotated PDF attached)

Line 27
"Infected" ->
Was it artificial infection? If not, "infected" could be deleted.

Line 38
"polyacrylamide gel electrophoresis"
-> "protein composition analysis by SDS polyacrylamide gel electrophoresis

Line 40
"ORF" - "ORF product"

Line 81
Chinese -> Asian

Line 134
"Fifty infected A. mellifera".
Were they just symptomatic or were artificially infected?

Line 138
"supernatant was then successively passed"
provide more details, supernatant after which centrifugation

Line 168
"Amplification of the full-length CSBV-SDLY genome" -> AmCSBV-SDLY genome sequencing"
I understood form materials and methods that no amplification of full genome (I.e. single 8.7 kip RT-PCR product) was carried out.

Line 183
"The amplified nucleotide fragments were assembled" -> "Nucleotide sequences of the amplified RT PCR fragments were assembled" (I understand no single cDNA fragment, 8 kbp full-length size was constructed.)

Line 184
"Multiple nucleotide ... " - indicate if sequence variation was reported between clones covering the same SBV genome regions.

Line 192
"A bootstrap value of 1,000 replicates was 193 applied to yield a robust phylogeny." -> "The phylogenetic tree was bootstrapped 1000 times and bootstrap values placed
on each branch (Fig. ?)"

Line 279
"was classified into a branch". -> "belonged to the clade"

Line 284
"Similarity analysis" -> "Similarity profile analysis"

Line 383
"the previously reported strains of CSBV" -> ? "the previously reported strains of CSBV which infect Apis cerana."

Line 384
"species barrier and cause death" -> "species barrier, infect and cause death..."

Line 386
"...between host populations" -> "between hosts

Validity of the findings

The paper reports detection in Apis mellifera of a CSBV strain closely related to a CSBV strain infecting A. cerana. The published sequence is not entirely novel, with 97% nt and 98% aa identity to one of previously sequenced CSBV strains. One of the paper novel findings relates to detection of recombination events between SBV strains. In this respect it is very important to provide additional data to demonstrate that all RT-PCR fragments sequences of which were assembled indeed derived from a single RNA genome.

Reviewer 2 ·

Basic reporting

.

Experimental design

.

Validity of the findings

.

Additional comments

This paper provides new data concerning the structure and variation of one isolate of the Sacbrood virus (SBV) obtained from Shandong, China (AmCSBV-SDLY). The isolate was obtained from Apis mellifera honeybees and completely sequenced, compared with other available sequences and a molecular phylogenetic tree was inferred.
The authors showed that, AmCSBV-SDLY clustered with other previously reported Chinese and Asian strains originating from A. cerana (AC genotype) but was separate from the SBV genomes from Europe originating from A. mellifera (AM genotype) and speculated that AmCSBV-SDLY could cross the species barrier and cause death in A. mellifera larvae based on the finding of the deletion mutation at the amino acid 369 residue 2,128.
The host species specificity for SBV has been discussed for a long time in a scientific literature. New data could enrich knowledge in this field. However, nucleotide and phylogenetic analysis of determined sequences was carried out insufficient to make definite conclusions. The experiments were conducting properly but it contains little new information. To my opinion, further analysis regarding virulence, biological properties of the viral strains, host specificity is needed. Therefore I would advise authors to complete their research and revise a manuscript.
In my opinion, the phylogenetic analysis is insufficient. Since the authors have sequenced only 1 complete SBV genome from A. mellifera species, they should sequence some more SBV genomes and analyze any genomic region in detail to estimate host species specificity, if it is possible. Besides, it is possible to carry out genetic analysis due to a special option of Mega 6 to estimate genetic diversity between different genetic groups, if they exist.
Detail comments are described below.
Method:
1. Table 1. Synthetic oligonucleotides for amplification of the CSBV genome
These primers do not have a large enough overlap (100-150 nucleotides) so when assembling the sequence using DNASTAR software, whether it ensures the continuous pairing without the gaps due to the perplexity at the two ends of the sequence during sequencing.
2. In the summary, the authors mentioned the use of MEGA 5.0 software to construct the phylogenetic tree however a MEGA 4.1 package was used in the method (line 190).
3. Line 101:
Sentence “ecent studies have shown that CSBV has no association with natural infections of A. mellifera” should be rewritten and add references for this.
4. Line 183:
The sentence: “The amplified nucleotide fragments were assembled to generate the entire genome of AmCSBV-SDLY using the DNASTAR program” is repeated of the upper part so should be deleted.
5. Line 237:
The sentence: “Multiple sequence comparisons showed that the sequence of AmCSBV-SDLY was similar to the sequences of the previously reported SBV/CSBV strains.” is repeated so should be deleted.
6. Line 241:
AmCSBV-SDLY was least similar (89.5% to 90.4% identity) to other SBVs but showed maximum similarity with CSBV-FZ-KM495267.1.
With how many percent similarity with CSBV-FZ-KM495267.1?
7. Line 374-380: ‘’However, AmCSBV-SDLY was isolated after CSBV-JL2014-KU574661.1, and the recombinant signals obtained from CSBV-FZ-KM495267.1, CSBV-JL2014-KU574661.1, and CSBV-GD AF469603.1 were detected by BootScan (Fig. 7); hence, we speculate that AmCSBV-SDLY may be a recombinant virus strain and its parent strain was derived from CSBV-FZ-KM495267.1, CSBV-JL2014-KU574661.1, and CSBV GD-AF469603.1’’ should be rewritten.
There is no evidence in the manuscript showing that AmCSBV-SDLY was isolated after CSBV-JL2014-KU574661.1.
8. References should be cited as specified of the journal in the text and the reference list.
For articles of the same author published in the same year, additional indicators such as a, b, and c are needed for follow-up (eg. Mingxiao et al., have 3 articles published in 2011 so please cite as Mingxiao et al., 2011a, b, c respectively).
9. The reference for Nguyen Thi Bich Nga and Le Thanh Hoa was published in 2013 but not 2016.
10. Line 137-138: ‘‘according to the method of Ma et al. (Mingxiao et al.,2011)’’. It is not sure whether Ma and Mingxiao are two different people or they are the same person? And please check throughout the manuscript.

---

## Round 0.2 · Minor Revisions

Please pay special attention to a number of corrections needed to make figures and tables more elucidative.

·

Basic reporting

The manuscript reports and analyses complete nucleotide sequence of Chinese Sacbrood virus CSBV strain AmCSBV-SDLY-2016. Although CSBV, a type of Scbrood virus (SBV), is common in Apis cerana, the reported AmCSBV-SDLY was detected in Apis mellifera. The sequence of AmCSBV-SDLY-2016 is very similar to other strains of CSBV/SBV with the higher nt identity (97%) being observed with such CSBV strains isolated from Apis cerana as CSBV-FZ, Genbank accession KM495267, and AcSBV-Viet1 Genbank KM884990).

Although AmCSBV-SDLY might not be the first CSBV type virus identified in Apis mellifera, - Fig. 1 shows that there are other sequences which belong to "Apis-cerana - CSBV clade) which were isolated from Apis mellifera (Fig. 1, sequences with prefix "Am-" AmSBV-V4-2014, AmSBV-Viet6-2014).

It is mentioned in Materials and methods that 359 colony-level samples were analysed, but it was not specified how many of these were Apis mellifera and Apis cerana (lines 131). Confidence intervals for the proportion of Apis mellifera colonies and Apis cerana colonies infected with CSBV should be calculated and included.

Table 4 (larval challenge) should be re-done to include description of treatments (Group description rater than number. Provide 95% confidence intervals for the proportion dead pupae and statistical significance of differences in mortality between groups for each repeat separately.



Other points to be addressed:


Line 35-37.
The nucleotide sequence was not identified by electron microscopy, SDS PAAG, or immunodiffusion assay. It is clear from Materials and methods (lines 219-228) that a series of RT-PCR fragments were generated using SBV/CSBV -specific primers (Table 3).


Line 130-134 and Table 1.
Analysis of "359 samples" should be moved to Results section. Specify how many Apis mellifera and Apis cerana samples were tested, calculate and provide the 95% confidence interval for a proportion of CSBV infected Apis mellifera ad Apis cerana colonies.


Line 221.
"genome was amplified by RT-PCR" -> "genome sequence was determined by Sanger sequencing of the RT-PCR fragments". Note that SBV genome was not amplified, instead several cDNA fragments corresponding to the sections of the CSBV genomic RNA were amplified by RT-PCR and their sequences were assembled in silico.

Fig. 1.
In the figure mark sequences which were sequenced in this study. Also specify hosts (A.cerana or A. mellifera) for the Genbank sequences. Include bootstrap values.


Line 271
"bystrand" -> "by strand"
No strand-specific qPCR analysis is presented and discussed in the section lines 271-279.

Lines 280-284
Is is possibly to distinguish serologically CSBV and SBV in the assay?


Lines 290-296 (Table 4, Larval Challenge)
Include description of the treatment groups in the table. Provide confidence intervals for the mortality proportions, provide statistical significance of the differences in mortality between groups.

Lines 322, 323.
Discuss in which parts of the CSBV polyprotein deletions were observed.


Lines 404-406
This study presents only isolate of AC type CSBV in Apis mellifera. This is not enough to claim that CSBV (AC type) "circulating in honey bee populations for a long time".


Line 438
The claim that AmCSBV-SDLY-2016 due to deletion at the codon 2128 is not substantiated.

Experimental design

Design is appropriate, but there are questions to reporting the results (see above).

Validity of the findings

See comments above

Reviewer 2 ·

Basic reporting

.

Experimental design

.

Validity of the findings

.

Additional comments

The authors have adaquately revised this paper and it is much improved. However, the paper still has some grammatical flaws and/or some typing errors that need to be checked carefully before publication.

---

## Round 0.3 · accepted · Accept

The manuscript has improved from the last round of review, thank you, very much for including the corrections asked for by the referees.